# Biofuels from Renewable Sources, a Potential Option for Biodiesel Production

**DOI:** 10.3390/bioengineering10010029

**Published:** 2022-12-25

**Authors:** Dhurba Neupane

**Affiliations:** Department of Biochemistry and Molecular Biology, University of Nevada, Reno, NV 89557, USA; dneupane@unr.edu; Tel.: +1-775-409-2265

**Keywords:** generation of biofuels, biodiesel, renewable resources, fossil fuel, population growth, greenhouse gas emissions

## Abstract

Ever-increasing population growth that demands more energy produces tremendous pressure on natural energy reserves such as coal and petroleum, causing their depletion. Climate prediction models predict that drought events will be more intense during the 21st century affecting agricultural productivity. The renewable energy needs in the global energy supply must stabilize surface temperature rise to 1.5 °C compared to pre-industrial values. To address the global climate issue and higher energy demand without depleting fossil reserves, growing bioenergy feedstock as the potential resource for biodiesel production could be a viable alternative. The interest in growing biofuels for biodiesel production has increased due to its potential benefits over fossil fuels and the flexibility of feedstocks. Therefore, this review article focuses on different biofuels and biomass resources for biodiesel production, their properties, procedure, factors affecting biodiesel production, different catalysts used, and greenhouse gas emissions from biodiesel production.

## 1. Introduction

The rising world population is predicted to reach over 9 billion by 2050 [1]. Increasing global prices and higher energy demand have put tremendous pressure on natural energy reserves, causing their depletion [2,3,4]. The burning of fossil fuels has several environmental implications, including an increase in greenhouse gas (GHG) emissions, particularly carbon dioxide (CO_2_) [5,6]. Over the last few decades, global primary energy consumption has increased dramatically due to rapid industrialization and higher living standards [2,7]. Developing countries such as Brazil, the South Asian region, and South Africa require 12–24 gigajoules (GJ)/cap of energy annually to have a decent standard of living [8]. Currently, over 80% of the world’s energy comes from fossil fuels, including natural gas, oil, and coal, and about 98% of it is generated via carbon emissions from fossil fuels [8,9]. The duration and intensity of drought are expected to become more severe, thus reducing water reserves by five-fold throughout the 21st century [1].

An increased share of renewable energy in the global energy supply will help to stabilize surface temperature rise to 1.5 °C compared to pre-industrial levels [10]. The temperature increase could be as much as 3–5 °C depending on certain regions [11]. Further, a shift in rainfall was found, ranging from 19.2 to 37.2 mm over different growing seasons [12]. With the inadequate pool of sources, particularly water, and an ever-increasing need for global energy, alternative fuels are the most practical way to meet the rising demand [13]. Researchers have already figured out alternatives to address this demand [14]. Further, the potential options to mitigate the effect of climate change and reduce dependence on fossil fuels are urgently needed and are already in development. There is an increasing interest in growing biofuels at a global and national level as a low-carbon alternative to fossil fuels due to their potential to reduce GHG emissions and the associated climate change impact from transport [15]. The use of bioenergy/biofuels is one of the promising renewable energy alternatives [16] because these are cheaper in synthesis [4]. Biofuels, generally biodiesel, have attracted researchers’ attention due to their potential benefits over fossil fuels and the flexibility of feedstocks. For example, sulfur-free, adequate oxygen content, an easy manufacturing process, and reduced GHG emissions are critical advantages of biodiesel [17]. Biodiesel as a diesel fuel is bio-degradable [18], non-toxic [19], portable, environmentally sustainable [20], efficient, and has low sulfur as well as aromatic content [21]. Additionally, due to the higher flask point of biodiesel, transportation and storage of biodiesel are safer than diesel fuels. However, it has some disadvantages; biodiesel is more expensive and emits more NO gas than diesel [9].

Due to its crucial characteristics and usage of versatile feedstock, for example, from waste frying oil to cheap non-edible resources, biodiesel has tremendous potential to use as an alternative fuel [22]. It is a promising and economical alternative to diesel that can reduce the global reliance on imported petroleum fuels. This article provides a comprehensive review of the types and generation of biofuels, biomass sources, properties, and factors affecting biodiesel production. This article also highlights various catalysts in biodiesel production, greenhouse gas emissions from several literatures, and finally, the conclusion and future perspective. 

## 2. Types and Generation of Biofuels

Biofuels are classified into four generations, namely first, second, third, and fourth based on their sources and production of various biomaterials. A brief description of each of the generations is highlighted below.

### 2.1. First-Generation Biofuels

First-generation biofuels are conventional biofuels, mainly generated from two types of edible feedstock, namely starch-based (e.g., potato, corn, barley, and wheat) and sugar-based (e.g., sugarcane and sugar beet) feedstocks [23,24]. The main advantages of first-generation raw materials are the availability of crops and comparative simple conversion processes. However, using edible food crops for biodiesel production, there is a reduced food supply, thus potentially increasing food prices [25]. Another concern is the diverting of agricultural land to fuel production. Using a significantly large amount of fertilizer and pesticides for agricultural production could negatively impact the environment [15]. There are several types of conventional biofuels based on the technological approach they use to generate (Figure 1). 

#### 2.1.1. Bio Alcohols

Bio alcohols are extracted with the help of enzymes and microorganisms by alcohol fermentation of cellulose, glucose, starches, carbohydrates, and other sugars. Bio alcohols are further categorized into bioethanol, biopropanol, and biobutanol [26]. 

#### 2.1.2. Biodiesels

Biodiesels are the forms of diesel extracted from renewable feedstocks, including lignocellulosic biomass, which consists of long-chain fatty acid esters. Biodiesels are produced chemically by reacting lipids, such as animal fat (tallow), soybean oil, or other vegetable oils with alcohol and produce methyl, ethyl, or propyl ester [27]. The commonly used catalyst used during biodiesel production includes NaOH or KOH [28].

#### 2.1.3. Vegetable Oil 

Vegetable oils are produced from fat, olive oil, castor oil, and sunflower oil. The fuels produced from vegetable oil are economical and environmentally friendly. Recent studies reported that waste cooking and vegetable oils are considered alternative fuels for diesel engines in some precise applications [29].

#### 2.1.4. Green Diesel 

The hydrotreating of triglycerides produces green diesel in vegetable oils with hydrogen. Three main reactions during the process are hydrodeoxygenation (HDO), decarbonylation (DCO), and decarboxylation (DCO_2_) [30].

#### 2.1.5. Biogas 

Biogas is produced by anaerobic digestion with the help of microbial consortium without oxygen, and digestate as a nutrient-rich byproduct is also produced [31,32]. Biogas produced during the process contains about 60% CH_4_, 35%CO_2_, and 5% a mixture of H_2_, N_2_, CO, NH_3_, O_2_, and volatile amines [33]. Biogas can be used for industrial energy, cooking in rural areas [33,34], and combined heat and power production [10]. 

#### 2.1.6. Solid Biofuels

Raw materials, including wood, wood chips, leaves, sawdust, charcoal, and animal dung, are commonly used as solid biofuels. The use of solid biofuels in the energy sector is limited to particular markets [23]. For example, firewood is the most common strategy to generate bioenergy, which can be used for cooking food [28]. 

### 2.2. Second-Generation Biofuels

The controversy of using first-generation biofuel feedstock due to the food vs. energy debate has forced us to move to second-generation biofuels, such as lignocellulosic or carbohydrate biomass, as the potential alternative source for biofuels and chemical production [24]. These feedstocks do not rely on edible plants and do not require agricultural land [35]. Cellulosic biomass comprises various chemical compositions such as cellulose, lignin, and polyose. Lignocellulosic biomass is composed of cellulose (35–50%), lignin (15–20%), hemicellulose (20–35%), and other components (15–20%). The lignocellulosic-based biofuel production process has the potential to lower GHG emissions, boost the economy, and aid energy security. The biotechnological approach in the United States has been estimated to produce 1.3 billion tons of dry biomass annually without compromising food security [36]. Second-generation biofuels are advanced biofuels obtained from several trees, grass, bushes, and agricultural residues [23]. Based on the technologies used to produce them, second-generation biofuels include the following (Figure 1). 

#### 2.2.1. Cellulosic Ethanol 

The fermented sugars obtained from polyose and cellulose compounds of lignocellulose are used for making cellulosic ethanol [23]. Cellulosic biofuels can contribute to rural economic development and enhance the sustainability of agricultural landscapes [37,38].

#### 2.2.2. Algae-Based Biofuels 

Algae is the fastest-growing raw material for biofuel production and an essential substitute for biofuel extraction. Techniques of extraction and concentration of biomass from algae include processes such as centrifugation, aggregation, floatation, purification, and flocculation [39,40]. Biofuels such as biodiesel, biogas, and hydrogen can be produced from algae using the advanced feature [41].

#### 2.2.3. Alcohol

Alcohol is obtained from syngas by fermenting biomass with the help of specific microorganisms [42]. 

#### 2.2.4. Dimethylfuran 

Dimethylfuran is an oxygenated hydrocarbon with an oxygen content of 17%. It is an additive in diesel fuels. This is highly competitive in reducing emissions from engines [43]. 

#### 2.2.5. Biosynthetic Natural Gas (Bio-SNG)

Biogas can be produced from anaerobic digestion with the help of microbes. Bio-SNG is used in the form of CNG and LNG in vehicles and for refilling a natural gas cylinder [44]. 

### 2.3. Third-Generation Biofuels

Third-generation biofuels are produced from algal biomass and waste oil. The advantages of using third-generation biofuels include higher growth and productivity, no agricultural land required, higher oil content, and less impact on food supply. Microalgae, fish oil, animal fat, and waste cooking oil are the primary sources of third-generation biodiesel feedstocks [45]. Because of the cost involved during harvesting, drying, and extraction processes, using algal biomass as biodiesel feedstock is expensive. However, it produces about 10–100 times more biofuel or oil per unit area. Seaweed or macro-algae is third-generation biomass that can be used in bio-energy production and has many advantages such as short cultivation time, high carbohydrate, proteins, and lipids content, and low or no lignin content [46]. Algal-based biofuel includes bioethanol, biodiesel, and biohydrogen (by the process of bio photolysis, photo fermentation, and dark fermentation) [47]. A study showed that the lipid in algae could be converted to biodiesel by the conventional approach, such as the conversion method used for vegetable oil. The conversion process of algal biodiesel production involves transesterifications, enzymatic, wet extraction, alcoholysis and acidolysis, and finally, biodiesel [48]. Algal oil blended with diesel fuel in a 20% ratio reduced hydrocarbon exhaust and better emission characteristics [49,50]; however, the complete combustion of algae releases a higher % of NO_x_ into the atmosphere due to the significant presence of nitrogen in algae (5–8%) [51].

In the case of waste oil or waste cooking oil, the variation in using different feedstocks and their chemical composition, and impurities, limit their productivity at large scale [52]. Waste coffee ground oils and bardawil lagoon are an example of third-generation feedstock used in recent years [53].

### 2.4. Fourth-Generation Biofuels

With the application of molecular biology, genetic engineering, and interdisciplinary physicochemical approaches, which include the use of CRISPR/Cas9 with guided RNA for genetic modification in algae [54] to optimize and enhance the yield of biofuel production, the biofuel generated by such process is considered a fourth-generation biofuel. The fourth-generation biofuel production employs genetically modified algae that accumulate high lipid and carbohydrate content to improve biofuel yield [55]. The raw materials used for biofuel production are microalgae, macroalgae, and cyno-bacteria. Cyno-bacteria are non-photosynthetic prokaryotes, and micro and macro algae are eukaryotes [56]. The inactivation of ADP-glucose phosphorylase in a *Chlamydomonas starchless* mutant led to a 10-fold increase in TAG [57]. Similarly, a modification in the CoA-dependent 1-butanol production pathway into a cyanobacterium, *Synechococcus elongatus*, can produce butanol from CO_2_ directly [58].

## 3. Biomass Sources for Biodiesel Production

Biodiesel or fatty acid methyl ester (FAME) is a processed diesel fuel from different biological sources, including edible, non-edible, animal fats, and waste cooking oils. FAME combines long-chain fatty acid monoalkyl esters of fatty acids [2]. It is a green biological ester-based oxygenated oil that comprises organic fats and oils [3]. The world’s biodiesel production is projected to reach 10.3 billion gallons by 2024, which reached 8.5 billion gallons in 2016 [59]. It is also estimated that food-based feedstocks (first-generation biofuel) will dominate the world’s market [60].

Biodiesel is intended to be used in standard diesel engines as a standalone fuel or blended with petroleum. In 2021, the total volume of biodiesel production in the United States amounted to over 1.6 billion gallons, compared to 9 million gallons in 2001 and 991 million in 2012. After 2012, there were fluctuations in biodiesel production volume in different years, with the highest quantity attained in 2018 (Figure 2a). Similarly, total biomass production in the United States was 1375.56 billion kW hours in 2021, which is expected to increase gradually in the coming decades. It is estimated to reach 1630.73 billion kW hours by 2050 (Figure 2b). 

Various feedstock sources can be used for biodiesel production, including various vegetable oils, animal fats, microbial oil, algal oils, and waste oils [63,64]. Palm oil, stearic oil, lauric oil, oleic oil, soybean oil, sunflower oil, palmitic oil, rapeseed oil, canola oil, and vegetable derivates are included under vegetable oils. With the use of catalyst, animal fats or vegetable with alcohols also produces biodiesel and glycerin [3]. Feedstock selection is a crucial step in biodiesel production, which is impacted by different factors, such as yield, cost, composition, and purity of the produced biodiesel. Another significant factor affecting biodiesel production is availability and the types of sources (non-edible, edible, or waste) [65].

Further, the choice of materials used for its production depends on the geographical regions; for example, soybean is the primary source of biodiesel in the United States, whereas, in Europe and the tropical parts of the world, rapeseed (canola) and palm oil serve as the primary sources [66,67,68]. Different feedstocks produce biodiesel with distinct qualities that must be considered when blending biodiesel with petroleum diesel for their use in transportation. Biodiesel is blended with petroleum diesel from 5% to 20% biodiesel, or B5-B20. However, the Renewable Fuel Standard (RFS), a federal program that mandates the blending of biofuels into the nation’s fuel supply, has suggested including higher biodiesel blends. Soybean and canola oil are the most common biodiesel in the United States. Soybean accounted for about 50% of biodiesel feedstock input between 2014 and 2017. The soybean oil used for biodiesel production increased by 30% in 2017 compared to 2014 (https://www.eia.gov/todayinenergy/detail.php?id=36052. Accessed on 3 May 2018). 

In 2020, approximately 71.7 % of the biodiesel feedstock came from soybean, while other small amounts of vegetable oils and animal fats (AF) such as canola oil (10.7 %), corn oil (13.0%), tallow beef fat (3.1%), poultry fat (1.5%), and other (0.1%) were used (Table 1). Based on 2012–2019 data (Table 2), rapeseed oil is still the dominant biodiesel feedstock in Europe and worldwide. In 2016, rapeseed (canola) input to global contribution for biodiesel production was 68%, followed by soybean (15%), animal fat and yellow grease (5% each), palm oil (6%), and sunflower (1%) [68]. However, rapeseed share in the feedstock mix in Europe has significantly decreased; for example, its share was 62.3% in 2012 compared to only 37.9% in 2019 (Table 2). This decrease in the share of rapeseed oil in Europe is primarily because of recycled vegetable oil/used cooking oil (UCO) and palm oil. UCO, or yellow grease, has become the second-most important feedstock for Europe since 2015. In the USA, biodiesel production from yellow grease (13%) dominated both rapeseed-based biodiesel (10%), corn-based biodiesel (12%), and animal fats-based biodiesel (10%) (based on 2016 data reported by Kim et al., 2018 [68]). 

The types of raw materials/feedstocks for biodiesel production rapidly diversified for economic and environmental reasons [69]. A market survey reported that biodiesel’s feedstock market is transitioning from first-generation feedstock such as soybean, rapeseed, and palm oil to non-food and lower-cost feedstock such as jatropha, castor, UCO, and AF [70]. In countries such as Brazil, effective programs are underway to promote jatropha and castor production for biodiesel production. Similarly, another emerging feedstock for Biodiesel is HVO, which is produced through hydrotreating [69]. Production and use of biofuel generate emissions such as particulate matter (PM), carbon monoxide (CO), carbon dioxide (CO_2_), nitrogen oxides (NOx), hydrocarbons, and volatile organic compounds (VOCs). The VOCs, unburnt hydrocarbon (UBHC), and NOx are the precursors for forming smog and ground-level ozone, which are associated with increased morbidity and mortality from cardiovascular and respiratory diseases and certain cancers [15]. Compared to fossil diesel, biodiesel produces lower PM, CO, VOCs, and NO_X_ emissions [71]. Among NOx, nitrous oxide (N_2_O) is only the greenhouse gas of great environmental concern. It is a substantial anthropogenic greenhouse gas, and agriculture represents its most significant source. The global warming potential of N_2_O is 298 times that of CO_2_ [72]. Previous studies on biofuel production systems revealed that emissions of N_2_O may counterbalance a substantial part of the global warming reduction by fossil fuel displacement [73]. Using optimized crop management, which involves state-of-the-art agricultural technologies coupled with an optimized fertilization regime, and nitrification inhibitors, N_2_O emissions can significantly be reduced by −135% points (pp) compared to conventional management. However, uncertainties in using statistical N_2_O emission models and data on non-land use GHG emissions due to biofuel production are significant, which can change the GHG emission reduction by between −152 and 87 pp [74].

While selecting the raw materials for biodiesel production, various parameters are considered, including oil content, suitability, chemical composition, and physical properties [75] (Table 3).

A brief description of various feedstocks used for biodiesel production with their oil content is summarized in Table 3. 

Different studies were conducted to investigate the suitability of various feedstocks, for example, edible, non-edible oils, animal fats, and algal oils, for biodiesel production. The transformation of edible oil is biodiesel was considered the most feasible approach. As reported in Table 3, the biodiesel feedstocks such as olive oil and microalgae oil have the highest oil content, up to 70%, followed by rubber seed oil (up to 68.4%) and coconut oil (up to 65%). The lowest oil content was reported for soybean oil (15–20%). Edible oils such as sunflower, soybean, and rapeseed (Table 3) served as important substrates for biodiesel production. However, a vast disparity in food use affects the use of these first-generation feedstocks as fuel [76]. This will create a significant conflict with food vs. fuel, and competition with the food market can also adversely affect the price of biodiesel. The shift for non-edible oil such as castor oil, jatropha oil, and rubber seed oil was associated with the higher price of biofuel from edible oils because of their higher demand for food. Using raw materials from non-edible oils, animal fats, and waste oils has several advantages, including reducing the price of raw materials and avoiding competition with the food market [25,64]. 

In recent years, there has been significant interest in renewable and sustainable oils, and the life cycle assessment of raw materials plays a vital role in biodiesel production, it is essential to consider the oil content (%) and oil yield to determine the quality of biodiesel [65]. Additionally, microalgae are a great source of biodiesel production. These organisms can produce well-graded bioactive compounds by converting carbon dioxide (CO_2_) with the help of sunlight [77,78]. With the increase in the price of petroleum and the concern with greenhouse gas emissions, microalgae have become an environmentally friendly alternative for biodiesel production. Though it is challenging for commercial-scale production, several companies have already started algal-based fuel production [77,78]. 

Similarly, animal fats, the byproducts of meat processing and cooking, are also important sources for biodiesel production. These include mutton or beef tallow, yellow grease, and lard, the residues after producing omega-3 fatty acids from fish oil [78]. Commercial-scale biodiesel production has been attained from animal fat-based feedstocks such as tallow, lard, and chicken fats. Unlike edible oils, animal fats-based biodiesel feedstocks have economic, environmental, and food security advantages. However, higher amounts of saturated fatty acids and free fatty animal fats demand complex production techniques. On the other hand, animal waste fats with lower saturated fatty acids have good oxidative stability, elevated calorific value, and shorter ignition [78,79]. Another important source of biodiesel feedstock is waste cooking oil. The waste cooking or frying oils include yellow and brown grease that does not directly conflict with food security. Yellow grease has < 15% fatty acid and can be used as a potential low-cost raw material for biodiesel production compared to brown grease (>15% fatty acid), which has an adverse effect on biodiesel production [79]. 

Feedstocks’ chemical composition and physical properties are essential when selecting raw materials for biodiesel production. The chemical composition of different fatty acids from different sources is highlighted in Table 4. The differences in the degree of saturation and the carbon chain length are mainly due to the fatty acids of different architecture in the oil [66]. The degree of saturation from different sources is 14.7 % (soybean oil), 49.6% (palm oil, 6.1% (rapeseed oil), 21.6% (jatropha oil), 28.7% (used cooking oil), 46.9 % (animal fats), and 36.1% (algal oil) [68]. The percentage of carbon found at higher concentrations with C ≥ 18 in most of the feedstock oils except for algal oil, which has only 33.1% compared to 85% (soybean oil), 55% (palm oil), 87.4% (rapeseed oil), 85.7% (jatropha oil), 73.1% (used cooking oil), and 68.9% (animal fats) [68]. This study compiled the fatty acid profile of different fatty acids from various sources, including edible and non-edible oil, animal fats, and other sources. The predominant fatty acids were monosaturated fatty acids, saturated fatty acids and polyunsaturated fatty acids, oleic acid (C18:1; 2.9–72.2), palmitic acid (C16:0; 1.3–48), and linoleic acid (C18:2; 1–70) (Table 4).

## 4. Biodiesel and Its Properties

Biodiesel, also known as FAME, is produced by mixing methanol with vegetable oil, animal fat, or other triacylglycerol-carrying material. Differences in feedstocks significantly fluctuate the value of characteristics of FAME, including cloud point, Cetane number (CN), oxidative stability, saponification value, iodine value, and acid value [88]. The main physicochemical properties of biodiesel obtained from various feedstock/raw materials are discussed below (Table 5). 

### 4.1. Cloud Point

The cloud point (CP) is the minimum temperature below which wax begins to form crystals in fuels, resulting in a cloudy appearance [98]. Solidified waxes can clog engine fuel filters and injectors. Biodiesel has higher CP due to the high melting points of saturated fatty acids compared to unsaturated fatty acids [88]. Biodiesel produced from feedstocks such as inedible tallow and waste frying oil may require additives or blend at higher levels with lower cloud point ULSD to mitigate cold weather concerns.

### 4.2. Cetane Number

The cetane number (CN) represents the ignition behavior and quality of the fuel. Higher cetane is often associated with improved performance and a cleaner burning fuel [99]. Most biodiesel feedstocks have slightly higher cetane numbers than ultra-low sulfur diesel (ULSD), which usually has a minimum allowable cetane value of 40 (https://www.eia.gov/todayinenergy/detail.php?id=36052. Accessed on 3 May 2018). The CN value of biodiesel increases with the length of the fatty-acid chain and the degree of saturation; hence, a higher CN means a higher oxygen concentration in the biodiesel and a better combustion efficiency [98]. Studies reported the highest CN value of 70 for Spirulina platensis [100] vs. the lowest CN value of 34.6 for biodiesel obtained from linseed oil [101,102]. The raw materials and feedstocks reported in this study have a CN value range between 47 for soybean oil and 64 for choice white grease (Table 5). 

### 4.3. Oxidative Stability 

Oxidative stability is the ability of the fuel to resist oxidation during storage and use. This essential factor significantly influences the storage duration and condition [103]. Fuels with lower oxidative stability are more likely to form peroxides, acids, and deposits that adversely affect the engine performance. Because it generally has lower oxidative stability, petroleum diesel can be stored longer than biodiesel feedstocks such as white grease and tallow. Biodiesel producers may use additives to extend the storage and usage timelines of Biodiesel (Source: EIA). Biodiesel with high oxidative stability is highly susceptible to oxidation deterioration. Oxidative stability varies according to fatty acid composition [104]. The fuel’s oxidative stability is greatly affected by polyunsaturated FAME. For example, Camelina-oil-based Biodiesel has low oxidative stability because it has approximately 35% polyunsaturated FAME occurrence (i.e., α -linolenic [C18:3]) [105] compared to coconut-oil-based biodiesel, which has better oxidative stability due to 2% polyunsaturated FAME in its oil [88]. Oxidative stability reported in this study ranges from 2.3 mg/100 mL (yellow grease) to 44.9 mg/100 mL (Canola oil) (Table 5).

### 4.4. Saponification Value 

Saponification value (SV) is an index of the molecular weights of triglycerides in the oil. It is inversely proportional to the average molecular weight or the chain length of the fatty acids [106]. Thus, the shorter the chain length, the higher the SV of the oil. The expected SV should range between 195 and 205 mg/KOH/g of oil [107,108]. Any value below that value needs refining to meet the required standard and would be better fitted for an industrial purpose [109]. The SV reported in this study is comparable and lies close to the required range of 195–205 (Table 5).

### 4.5. Iodine Number

Iodine number (IN) represents the amount of iodine absorbed by double bonds of the FAME molecules in 100 g of the fuel sample. A higher iodine value indicates higher fats and oils [110,111]. In the case of biodiesel fuels, linseed methyl ester showed the highest IN of 178 compared to the lowest IN of 37.59 reported for Kusum-oil-based Biodiesel [102,112]. This study reported the lowest IN for Palm oil (35–61) vs. the highest value of IN for Camelina oil (146.5) (Table 5).

### 4.6. Acid Value

The acid value represents the fuel sample’s quantity of free fatty acids. A high acid number causes corrosion problems in the engine’s fuel delivery system [112]. A high acid value of 6.9–50.8 mg KOH/mg of oil is reported for biodiesel from palm oil compared to the lowest acid value of 0.1–0.2 mg KOH/mg of oil from soybean oil (Table 5). Further, descriptions of additional fuel properties of biodiesel from different generation oil feedstocks are reported in our previous study [88].

## 5. Procedures for Biodiesel Production

Different physicochemical processes could produce biodiesel, and the primary methods include pyrolysis, micro-emulsion, and transesterification [113]. Each method has its merits and demerits. For example, micro-emulsion is a simple and environmentally safer method that generates fewer pollutants. Biodiesel synthesized using this method has a good cetane number (CN). Similarly, alcohol in the micro-emulsion process improves the CN of Biodiesel [114]. Microemulsion-based fuel systems reduce the combustion temperature, which leads to lower emissions of thermal NO_x_, CO, black smoke, and particulate matter. However, one major problem of using ethanol to formulate a microemulsion system is its lower miscibility with diesel. The immiscibility can be visualized for a wide range of temperatures, particularly at lower temperatures [115,116]. Furthermore, environmentally benign bio-based non-ionic surfactants and cosurfactant without N and S are of environmental concern [117,118]. Biodiesel production from the pyrolysis method (also known as thermal cracking) has low CN, volatility, and high viscosity [21]. By comparing these methods, the transesterification method is reliable and effective because the transesterification method demands low temperature, low pressure, and less processing time. The transesterification method is simple and highly efficient [119]. A description of various procedures to generate biodiesel is highlighted below. 

### 5.1. Micro-Emulsion

This method uses isotropic fluid to form a colloidal dispersion of dimensions ranging from 1 to 150 nm. A study using soybean oil has already demonstrated that by using this method, maximum viscosity was achieved that involves both ionic and non-ionic aqueous solutions [120,121]. A study revealed that using a ternary phase system (a clear and thermodynamically stable, isotropic liquid mixture of oil, water, and surfactant) counters the viscosity problems of vegetable oils by forming micro-emulsions with different solvents (ethanol, methanol, propanol, n-butanol, and hexanol). These alcohols act as emulsifying agents, dispersing the oil into tiny droplets, usually with diameters ranging from 100 to 1000 Å [122].

### 5.2. Pyrolysis

Thermal cracking or pyrolysis converts organic materials to fuels without oxygen using thermal decomposition (temperature: 300–1300 °C) [122]. Chemically, pyrolysis reaction cleaves the bonds in a substance, converting it into many smaller compounds. The process is similar to the process used to synthesize petroleum-diesel; therefore, it yields a product with similar combustion characteristics and results in less waste formation and no pollution [123,124].

The substrate used for pyrolysis includes vegetable oils, animal fats, natural fatty acids, or methyl esters of fatty acids. It sometimes produces a higher yield than the transesterification reaction, which is the most widely used [122]. The pyrolysis of organic feedstock for the manufacture of synthetic diesel has yet to be viable on an economic scale [124]. Based on operating parameters, pyrolysis can be divided into three types, namely conventional pyrolysis (550–900 K), fast pyrolysis (850–1250 K), and flash pyrolysis (1050–1300 K) [124]. The pyrolysis of biomass for bio-oil generation can be performed using both conventional and flash pyrolysis. In conventional pyrolysis, the vapor residence time ranges from 5 to 30 min, and thus this contributes to overall reaction time. Depending upon residence time, the vapors can be removed continuously. Whereas in flash pyrolysis, the heating rate is predominantly high. Some of the prerequisites for flash pyrolysis include a high heat transfer rate, finely grounded materials, and short vapor residence times (<2 s) [125]. The product obtained from pyrolysis has desired characteristics of biodiesel, such as low viscosity, less amount of sulfur and water, and high cetene number; however, it has less ash and residual carbon content than the desirable amount [123,124,126,127]. 

### 5.3. Transesterification

Transesterification is a standard and widely used procedure for high-quality biodiesel production [128]. This procedure involves the transformation of fats or oils using alcohol, particularly methanol or ethanol, with the help of catalysts (e.g., heterogeneous, homogeneous, or enzyme) [129,130]. Compared to the transesterification process facilitated by enzymes, the process is energy-consuming because of the presence of soap byproducts, and separation and purification of the chemically produced biodiesel require more complex steps than enzymatically produced biodiesel [131]. Ethanol is cost-effective and abundant commodity obtained from the fermentation of sucrose from sugarcane. Propanol or butanol could be a better option because these two alcohols promote better miscibility between the alcohol and the oil phases [132]. Transesterification can be combined with ultrasound-assisted member technology [25,66,120]. There are merits and demerits of using various biodiesel production technologies based on several studies (Table 6). However, these production technologies were centered on reducing problems during biodiesel production, such as oil’s high viscosity, acid value, and fatty acid content [97,133]. Among those technologies, transesterification using a homogeneous catalyst was the most typical and commercially used technology [134]. From an environmental point of view, enzyme catalysts and heterogeneous catalysts are suitable options for the future [75]. 

## 6. Factors Affecting Biodiesel Production

Biodiesel production using biomass feedstock is influenced by several factors described below. 

### 6.1. Free Fatty Acids

Free fatty acids affect biodiesel production. The higher amount of free fatty acid leads to soap and water formation [146]. The slow rate of acid-catalyzed reaction requires low-temperature conditions [147]. Base-catalyzed transesterification reactions demand raw materials with low acid value (<1) and free from water [148]. With 3% free fatty acids, there is no need to use a homogeneous base catalyst during the transesterification reaction [149]. 

### 6.2. Water Content

The amount of water content in the feedstock accelerates the hydrolysis and lowers the formation of ester [150]. A study has revealed that for a 90% biodiesel yield for an acid-catalyzed reaction, the water content should be less than 0.5% [151]. Additionally, water obtained as a byproduct inhibits the reaction and decreases engine performance. However, water in oil can be removed by preheating it up to 120 °C or by using anhydrous sodium sulfate or anhydrous magnesium sulfate [152]. 

### 6.3. Types of Alcohol 

Methanol is used for biodiesel production for a higher conversion rate from waste cooking oil with lower viscosity and is cheaper than other alcohol-based biofuels [153]. However, it is more toxic [154] and causes enzyme deactivation, denaturation, or inhibition at higher concentrations [155]. In order to address these issues, ethanol is used in most enzymatic reactions [153].

### 6.4. Alcohol to Oil Ratio

In order to obtain one mole of alkyl ester, 3 mol of alcohol and 1 mol of triglyceride are needed [156]. The rate of biodiesel production increases with higher alcohol concentration, i.e., increasing the alcohol-to-oil ratio [157]. The maximum conversion with 99% biodiesel production was achieved from waste sunflower oil transesterification using methanol and NaOH as the catalyst, with an alcohol-to-oil ratio of 6:1 [158,159], compared to 49.5% yield in waste canola petroleum using 1:1 methanol to oil [158]. 

### 6.5. Reaction Time

Reaction time plays a significant role in product conversion. Suppose more time is needed to give to the reaction. In that case, some parts of the oil may remain unreacted and ultimately reduce ester yield and exceed reaction time than usual, affecting the end product and leading to soap formation [160]. The reaction time for lipase-catalyzed reactions differs from 7 to 48 h [161]. Studies also suggested that reaction time also controls production costs. A study found no significant change in the conversion of biodiesel with the reaction time of 1 h (96.10%) versus 3 h (96.35%) [162]. However, a longer reaction time may lead to the reduction in biodiesel due to reversible transesterification reaction resulting in loss of esters and soil formation. Thus, reaction time needs to be optimized to bring the production cost down to a minimum. Maximum ester conversion can be achieved within <90 min.

### 6.6. Reaction Temperature

High temperatures lead to lower oil viscosity, resulting in a high reaction rate and reduced reaction time. However, if the temperature increases beyond the desirable range, the biodiesel yield is lowered due to the saponification of triglycerides accelerated by high temperature [163]. Biodiesel viscosity improves as the reaction temperature falls below 50 °C. For waste cooking oil, it is necessary to pre-heat up to 120 °C and cool down to 60 °C [164]. Higher reaction temperature increased the reaction rate and shortened the reaction time due to the reduction in the viscosity of oils. For the esterification reaction, the temperature should be below the boiling point of alcohol to prevent alcohol evaporation [165,166]. The highest conversion was achieved for cottonseed oil at 50 °C and Jatropha oil at 55 °C using lipase as a catalyst [167]. Further, the maximum yield of biodiesel was reported at 65 °C for domestic and commercial (waste and fresh) oils using KOH as a catalyst [162].

### 6.7. pH

Though pH is not crucial for acid/base catalysts, for lipase catalysts, pH plays an important role; for example, the enzyme may decompose at higher or lower pH. For example, a study found that a pH of 7 is optimal for biodiesel production using Jatropha oil-immobilized *Pseudomonas fluorescence* [168].

### 6.8. Catalyst Concentration 

The most commonly used catalyst for biodiesel production is sodium hydroxide (NaOH) or potassium hydroxide (KOH) [165], and other catalysts used are sodium methoxy and potassium methoxide [169]. Increasing the catalyst concentration with oil samples also increases the conversion of triglycerides into biodiesel. However, it also increased soap formation. Lowering the amount of catalyst leads to incomplete conversion into fatty acid ester, resulting in lower methyl esters yield [166]. Optimum biodiesel production is achieved when the concentration of NaOH reaches 1.5% weight [93]. Again, using an excess amount of catalyst can have a negative impact on biodiesel yield [93,170]. For soybean oil biodiesel, a 1.5 % copper vanadium phosphate (CuVOP) concentration was found to be the most effective [171].

### 6.9. Agitation Speed

Agitation is mandatory for the reaction, and its speed is essential for product formation. Lower agitation speed cause less product formation. Lower agitation speed cause less product formation. However, higher agitation speed favors soap formation [166]. There should be an optimum stirrer speed, which varies with our feedstocks. A study revealed a stirrer speed of 200 mm found to be optimum for biodiesel production using enzymatic reactions [172]. However, another study reported that at 400 rpm, there was a higher conversion of end product compared to 200, 600, and 800 rpm for an hour [165]. 

## 7. Catalyst Use for Biodiesel Production

Biodiesel is fatty acid methyl esters (FAME) with lower alkyl esters and long-chain fatty acids. It is synthesized by two procedures: esterification of fatty acids and transesterification with lower alcohol. Even without a catalyst, transesterification reactions can happen. However, they demand high temperatures, pressure, and reaction time. It also increases the overall cost of the reaction process [173]. The biodiesel thus produced has high purity of ester and glycerol (soap-free); however, from a commercial scale standpoint, it is imperative to use catalysts. Hence, there are three different catalysts: acidic, alkaline, and enzyme [174].

### 7.1. Acidic Catalysts

Acidic catalysts support higher efficiency for the esterification of FFAs over alkaline catalysts, with up to 90% conversion [175]. These catalysts favor feed oil with high acid value (including edible waste oil) and have good potential for transesterifying low-quality feeds [3]. Transesterification is performed at high temperatures (100 °C), pressure (~5 bar), and a high amount of alcohol. However, the process is slower compared to alkaline catalysts [3]. The most commonly used acid catalysts are sulfuric acid, hydrochloric acid, organic sulfonic acid, sulfonic acid, and ferric sulfate [75]. 

### 7.2. Alkaline Catalysts

Alkaline catalysts for biodiesel production include NaOH, KOH, alkaline metal carbonate, sodium and potassium carbonates, sodium methoxide, and sodium ethoxide. These catalysts are appropriate for oil with low FFAs due to the sensitivity as oils with higher FFAs, are converted to soap rather than biodiesel. This process restricts the separation of glycerin, biodiesel, and water. In order to cope with the issue, a deacidification step is necessary before the transesterification of vegetable oil [3]. 

### 7.3. Enzyme Catalysts

Enzymes such as lipases from microorganisms act as catalysts during transesterification reactions [176]. Lipase enzymes are abundant in nature and are synthesized by microorganisms (fungi, bacteria, and yeast), plants (rapeseed, oat, papaya, latex, and caster seeds), and animals (cattle, pigs, hogs, and pancreases of sheep) [177]. During biodiesel production, no or little residual or soap is formed at the end, resulting in high-quality glycerol production. This is also useful for feedstocks with high acidic values. Some limitations of using enzyme catalysts are high concentration and long reaction time. Separating the final product from the reaction results in a high cost of biodiesel production [3]. Further, applying metagenomics in enzyme technology opens the door for developing stable and solvent-tolerant biocatalysts for biodiesel production [178].

### 7.4. Homogeneous Catalysts

Homogeneous catalysis involves a series of reactions involving a catalyst from the same phase as the reactants, whether in the liquid or gaseous state. A homogeneous catalyst is dissolved or co-dissolved in the solvent with all the reactants [166]. Sodium hydroxide (NaOH) or potassium hydroxide (KOH) is the most popular homogeneous catalyst for biodiesel production [179]. Homogeneous catalysts are acidic and basic and widely used for biodiesel production. Acid catalysts are less active than base catalysts (i.e., lower reaction time). Therefore, a base catalyst involves high temperature and pressure. When FFAs exceed 1% in the oil, acid catalysts become effective. Acid catalysts prevent soap from forming. These catalysts catalyze the esterification of FFAs to form FAME and thus enhance biodiesel production [75,180]. The deep eutectic solvents (DESs) with acidic nature were evaluated for biodiesel production and found to have over 90% conversion efficiency [3]. Alkaline catalysts react with alcohol to form alkoxide and protonated catalysts. The carbonyl atom of the triglyceride molecule is attacked by nucleophilic alkoxide to form a tetrahedral intermediate, which reacts with alcohol to revive the anion. Further, the tetrahedral structure undergoes structural reorganization to form a fatty acid ester and diglyceride [66,181]. The higher conversion rate is obtained at low temperatures and pressure, resulting in lower production costs of biodiesel [3]. The alkaline catalysts are less efficient than acidic catalysts for converting oils containing high FFAs, producing soap, and inhibiting the separation of ester and glycerin. Thus, acid catalysts are recommended for biodiesel production [75].

### 7.5. Heterogeneous Catalysts

Catalysts with a state or phase different from reactants are heterogeneous catalysts. Most of the heterogeneous catalysts are solid. However, reactants are either in liquid or gaseous forms [166]. The separation process in heterogenous catalysts is easy and aids faster recycling and reuse than homogeneous catalysts. Therefore, it resolves problems related to homogeneous catalysis while lowering the material and processing costs [25,120,182,183,184,185]. Heterogenous catalysts can also tolerate high FFA and moisture content [186]. These catalysts, even at severe reactions conditions, can recover from a reaction mixture, stand up to aqueous treatment steps, and can be easily modified to achieve a high level of activity, selectivity, and long lifetime. Solid base heterogeneous catalysts include hydrotalcite, metal oxides (CaO, MgO, or SrO), oxides of mixed metals (Ca/Mg, Ca/Zn), alkali metal oxides (Na/NaOH/y-Al_2_O_3_, K_2_CO_3_/Al_2_O_3_, magnetic composites, and alkali-doped metal oxides (MgO/Al_2_O_3_, CaO/Al_2_O_3_, Li/CaO) [187]. However, some limitations of using heterogenous catalysts include diffusion due to phase separation between alcohol and oil, low surface area, and leaching. Strategies to resolve these issues include using n-hexane and tetrahydrofuran as co-solvents, increasing the area of specific activities, and providing more pores for reactive components. This can be possible with the help of supporters for catalysts as well as promoters for its structure [25,120]. The study also suggested that through immobilization or in the liquid phase, higher biodiesel yield can be obtained with robust lipase enzymes (how lipase technology contributes to the evolution of biodiesel production using multiple feedstocks).

A clear distinction between acid versus alkali and homogeneous versus heterogeneous catalyzed transesterification reactions is shown in Table 7 and Table 8.

Overall, the reusability and recyclability are complex in homogeneous catalysts, whereas heterogeneous catalysts offer efficient, yielding results and can be reused again. Nanocatalysts that come under the heterogeneous catalyst group reveal better yield due to the large surface area at the Nanoscale and are preferable for biofuel reproduction with the help of transesterification [4]. Homogeneous catalysts are also considered fuel performance catalysts due to their ability to improve fuel efficiency and reduce smoke emissions, unburned hydrocarbons, and carbon monoxide.

## 8. Evaluation of Greenhouse Gas Emissions from Biodiesel Production

Overall, biodiesel reduces GHG emissions of carbon monoxide (CO), carbon dioxide (CO_2_), unburnt hydrocarbon (UBHC), and particulate matter (PM) to a significant extent, except nitrogen oxide (NO_X_), compared to diesel. These emission gases are the primary causes of atmospheric pollution and human health [71]. 

Carbon monoxide reduction from different biodiesel feedstock range from 9.4% (microalgae) to 63% (palm oil) compared to CO emissions from diesel [188]. Several studies have shown the different proportions of CO emission reduction as engine speed increases. For example, CO emission reduction for soybean biodiesel was reported at 14% at 1400 rpm, 27% at 2000 rpm [189], and 37% at 3600 rpm engine speed [90]. Carbon monoxide reduction using rapeseed oil was 29.7% [91] and 26% [90]. Similarly, CO emission from jatropha oil ranged from 14 to 30% [190,191,192], waste cooking oil ranged from 17.8 to 20% [91,193], animal fats from 26% [90], and microalgae ranged from 9.4 to 32% [194,195,196]. 

Carbon dioxide emissions in some biodiesels are almost the same or even higher than the regular diesel [68]. For example, CO_2_ emissions from soybean oil (SO) and used/waste cooking oil (UCO)-based biodiesel were 60% and 33% more compared to regular diesel [90]. However, another study reported an increase in CO_2_ emission from SO and UCO by 1.8% and 1.2%, respectively. Similarly, palm, rapeseed oil [90], and jatropha oil [191] generated 41%, 32%, and 3% more CO_2_ emissions than regular diesel. Animal fats and microalgae biodiesels emitted 3% and 2.6% more CO_2_ emissions than regular diesel [197]. 

Nitrogen oxide emissions from biodiesels are more than diesel, except for palm oil and microalgae-based biodiesel. Studies have shown that biodiesel with long-chain fatty acids produced fewer NOx emissions than short-chain fatty acids. On the contrary, NOx emissions increased as the number of double bonds, i.e., the degree of saturation of fatty acids, increased [68]. NOx emissions from soybean biodiesel increased due to its highest degree of saturation (14.7%) and 85.3% of the chain. NOx emission was lowered for palm-based biodiesel due to more short chains and a high degree of saturation than other biodiesels [68]. 

PM emissions from biodiesel are lowered compared to diesel. PM emissions from SO biodiesel decreased from 56% [90] to 69% [198], palm oil by 50% [91], rapeseed oil by 36% [90] to 70% [198], jatropha oil by 11% [191] to 15% [192], and used/waste cooking oil by 17% [90]. Similarly, PM emissions from animal fats-based biodiesel decreased by 61% [99] to 77% [198] and microalgae by 31% [196]. 

Overall, soybean and animal fats-based biodiesel produced the lowest PM, jatropha oil-, animal fats-, and microalgae-based biodiesel produced the lowest CO_2_ emissions. Palm oil-based biodiesel produced the lowest CO and NOx emissions [68]. 

## 9. Conclusions and Recommendations

The demands for fossil fuels are gradually increasing due to the improvement in technology (for example, urbanization and improved life standard), which also requires more fuels. This increasing use of energy reserves will decrease fossil fuels in the future. A rapid increase in population and associated energy demand cannot be fulfilled by using fossil fuels alone. Using first-generation crops such as soybean and corn as bioenergy creates conflict in the food versus energy debate. Likewise, second-generation crops, particularly grasses, are unsuitable for biodiesel production. One of the significant problems in using second-generation vegetable oil is that it lessens engine life if the oil is not refined correctly. These issues of using first- and second-generation biofuels, such as economic, social, and food insecurity [48], can be resolved using third and fourth-generation biofuels. Third and fourth-generation biofuels are generated from various types of algae, which is highly efficient, and algal-based biofuels have great potential and no competition for food or land. In recent times, fourth-generation biofuels have great promise to overcome the inherent flaws and meet the world’s growing energy demands. Though algal cultivation is simple, feedstock production is complex due to high lipid content, and harvesting needs should be addressed. Detailed work on the parameters for fuel compatibility is required. Many things need to be worked out to make an algal biofuel a commercially viable option to fossil fuel, as the production of biofuels from microalgae is an energy-intensive process [199]. Further, greenhouse gas emissions are much lower; mainly, there is no emission of CO or CO_2_ using this generation of biofuels. Thus, these fuels could be potential options to replace fossil fuels. It is also recommended to consider the potential benefits of using other resources for energy sources that are more cost-effective, climate resilient, and sustainable. This could reduce the burden on fossil fuels in the future.

## Figures and Tables

**Figure 1 bioengineering-10-00029-f001:**
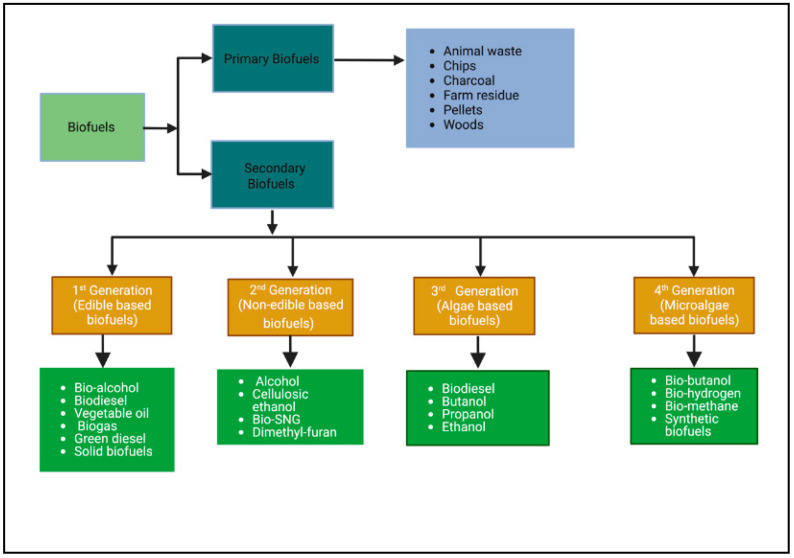
Types and generations of Biofuel. Adopted from [23].

**Figure 2 bioengineering-10-00029-f002:**
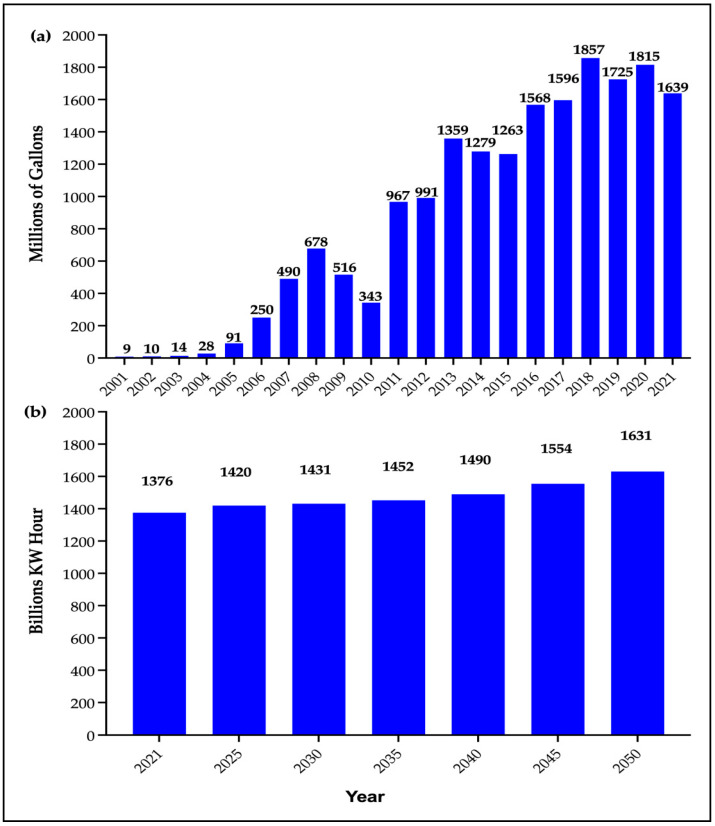
(**a**) US biodiesel production change for the past 20 years since 2001 [61] (Source: U.S. Energy Information Administration (EIA), Monthly Energy Review, Table, 10.4. Release date: April 2022. Available at: https://www.eia.gov/totalenergy/data/monthly/pdf/mer.pdf), and (**b**) US biomass energy production forecast from 2021 to 2050 [62] (Source: EIA, Annual Energy Outlook 2022, Table 1. Available online: https://www.statista.com/statistics/264029/us-biomass-energy-production/. Accessed on 21 June 2022).

**Table 1 bioengineering-10-00029-t001:** US inputs to biodiesel production (million kilograms).

Period	Vegetable Oil (Million kg)					Animal Fats (Million kg)	
	Canola Oil	Corn Oil	Cottonseed Oil	Soybean Oil	Other	Poultry	Tallow
January	49.4	80.3	-	236.3	W	5.0	W
February	42.4	60.7	-	260.7	W	5.4	9.4
March	59.6	65.7	-	297.6	W	10.7	W
April	62.8	38.0	-	304.7	S	W	10.9
May	58.9	38.3	-	365.3	W	3.9	5.3
June	50.0	42.7	W	338.9	5.9	W	9.7
July	W	60.5	W	351.5	W	W	24.6
August	W	67.3	W	338.0	W	W	20.0
September	W	61.7	-	334.0	W	10.4	12.4
October	W	45.8	-	328.0	W	9.5	23.6
November	W	60.3	-	309.8	-	6.4	15.0
December	W	66.7	-	337.5	-	3.2	17.2
Total	565.2	687.6	0.3	3802.5	W	78.5	166.9
% of total	10.7	13.0	0.0	71.7	0.1	1.5	3.1

Table with -, W, S indicates no data, withheld to avoid disclosure of individual company data, and value is less than 0.5 of the table metrics. However, the value is included in any associated total. Source: U.S. Energy Information Administration (EIA), Form EIA-22M “Monthly Biodiesel Production Survey.” U.S. EIA| Monthly Biodiesel Production Report (2020).

**Table 2 bioengineering-10-00029-t002:** The feedstock was used for biodiesel + renewable diesel (HVO; hydrotreated vegetable oil) in Europe from 2012 to 2019.

Feedstocks	2012	2013	2014	2015	2016	2017	2018	2019
Rapeseed oil	6500	5710	6200	6400	6060	6300	5200	5000
Used cooking oil (UCO)	800	1150	1890	2400	2620	2770	2860	2750
Palm oil	1535	2340	2240	2340	2315	2650	2570	2640
Soybean oil	720	870	840	540	610	930	1000	1100
Animal fats	360	420	920	1030	795	795	800	800
Sunflower oil	300	290	310	210	250	180	185	190
other, pine/tall oils, fatty acid	220	335	370	560	615	635	680	700
Share of rapeseed oil (%)	62.3	51.4	48.6	47.5	45.7	44.2	39.1	37.9

The original data were collected in a metric ton (MT) and then converted to kilogram (kg) using a conversion rate of 1 MT = 1000 kg (Source: EU-28. Available online: https://apps.fas.usda.gov/newgainapi/api/report/downloadreportbyfilename?filename=Biofuels%20Annual_The%20Hague_EU-28_7-15-2019.pdf. Accessed on 15 July 2019).

**Table 3 bioengineering-10-00029-t003:** Different sources of feedstocks/raw materials are used for the production of biodiesel [75].

Edible Oils	Oil Content (%)	Non-Edible Oils	Oil Content (%)	Animal Fats and Other Sources	Oil Contents (%)
Sunflower oil	25–35	^1^ Jatropha oil	30–60	Mutton fat	-
Soybean oil	15–20	Stillingia oil	44.15	Broiler chicken waste	41 [80]
Rapeseed oil	38–46	^1^ Karanja oil	27–40	Algae oil	20–60 [81]
Peanut oil	45–55	Neem oil	20–30	Waste cooking oil	33–53 [82]
Palm oil	30–60	^1^ Castor oil	45–60	Microbial oil	23–70 [83]
Olive oil	45–70	Rubber seed oil	53.7–68.4	Waste fish oil	40–65 [84]
Mustard oil	40–42 [85]	^1^ Mahua	35–40	Microalgae	30–70, 15–77
^1^ Linseed oil	35–45	-	-	Pine and Kapok oil	-
Coconut oil	63–65	-	-	-	-
Canola oil	40–45	-	-	-	-

^1^ represents feedstocks for biodiesel production reported by [86,87,88]; Ambat et al., 2018 [75] gathered information on sources of biodiesel feedstocks from different studies and reported them in their review paper.

**Table 4 bioengineering-10-00029-t004:** Fatty acid composition (%) of different biodiesel feedstocks.

Fatty Acid	Octanoic C8:0	Decanoic C10:0	Lauric C12:0	Myristic C14:0	Palmitic C16:0	Palmitoleic C16:1	Stearic C18:0	Oleic C18:1	Linoleic C18:2	Linolenic C18:3	Arachidic C20:0	Eicosenoic C20:1	Eicosapentaenoic C20:5	Behenaic C22:0	Erucic C22:1	others
Edible																
Soybean				0.1 ^a^	6–11 ^abc^	11 ^a^	2–5 ^abc^	20–30 ^abc^	50–60 ^abc^	5–11 ^abc^						
Rapeseed					1–3.5 ^bc^	9.1 ^c^	0–1 ^bc^	10–15 ^b^, 64.1 ^c^	12–15 ^b^, 22.3 ^c^	8–12 ^b^, 0.1 ^c^		7–10 ^b^			45–60 ^b^	
Sunflower					5–8 ^b^		2–6 ^ab^	15–40 ^ab^	30–70 ^ab^	3–5 ^b^	0.3 ^a^					
Peanut					8–9 ^b^		2–3 ^b^	50–65 ^b^	20–30 ^b^							
Olive					9–10 ^b^		2–3 ^b^	72–85 ^b^	10–12 ^b^	0–1 ^b^						
Palm				16.3 ^a^, 0.5–2 ^b^	8.4 ^a^, 39–48 ^bc^		2.4–6 ^abc^	15.4 ^a^, 36–44 ^bc^	2.4 ^a^, 9–12 ^bc^		0.1 ^a^					
Mustard							1–2 ^b^	8–23 ^b^	10–24 ^b^	8–18 ^b^		5–13 ^b^			20–50 ^b^	
Coconut			45–53 ^b^	16–21 ^b^	7–10 ^b^		2–4 ^b^	5–10 ^b^	1–2.5 ^b^							
Almond kernel					6.5 ^e^		1.4 ^e^	70.7 ^e^	20 ^e^	0.9 ^e^						
Walnut kernel					7.2 ^e^		1.9 ^e^	18.5 ^e^	56 ^e^	16.2 ^e^						
Sesame					13 ^e^		4 ^e^	53 ^e^	30 ^e^							
Non-edible																
Linseed					4–7 ^b^		2–4 ^b^	25–40 ^b^	35–40 ^b^	25–60 ^b^						
Neem					13.6–16.2 ^b^			49.1–61.9 ^b^								
Jatropha				0–0.1 ^a^, 14.1–15.3 ^b^	14.1–15.3 ^ac^, 0–13 ^b^	0–1.3 ^a^	3.7–9.8 ^ac^	34.3–45.8 ^abc^	14.1–15.3 ^b^, 29–44.2 ^ac^	0–0.3 ^ab^	0–0.3 ^a^			0–0.2 ^a^		1.4
Cotton seed					23–28.3 ^b^		0.8–0.9 ^b^	13.3–18.3 ^b^		0.2 ^b^						
Rubber				2.2 ^f^	10.2 ^f^		8.7 ^f^	24.6 ^f^	39.6 ^f^	16.3 ^f^						
Karanja					9.8 ^a^, 3.7–7.9 ^f^		2.4–8.6 ^af^	44.5–72.2 ^af^	10.8–18.3 ^af^							
Pongamia				11.65 ^f^				51.5 ^f^	11.65 ^f^							
Stillingia			0.4 ^f^	0.1 ^f^	7.5 ^f^		2.3 ^f^	16.7 ^f^	31.5 ^f^	41.5 ^f^						
Animal fat and other sources																
Animal fats				2.52 ^c^	28.4 ^c^		15.7 ^c^	42.2 ^c^	9.4 ^c^	0.6 ^c^	0.16 ^c^	0.86 ^c^		0.01 ^c^	0.01 ^c^	
Chicken fats				3.1 ^g^	19.82 ^g^		3.06 ^g^	37.62 ^g^								
Used/waste cooking oil				0.9 ^c^	20.4 ^c^, 8.5 ^g^	4.6 ^c^	4.8 ^c^, 3.1 ^g^	52.9 ^c^, 21.2 ^g^	13.5 ^c^, 55.2 ^g^	0.8 ^c^, 5.9 ^g^	0.12 ^c^	0.84 ^c^		0.03 ^c^	0.07 ^c^	0.04 ^c^
Tallow				23.3 ^f^	19.3 ^f^		42.4 ^f^	2.9 ^f^	0.9 ^f^	2.9 ^f^						
Brown grease				1.66 ^f^	22.83 ^f^		12.54 ^f^	42.36 ^f^	12.09 ^f^	0.82 ^f^						
Microalgal	0.2 ^d^			12–15 ^g^	34.8 ^d^, 10–20 ^g^	32 ^d^	1.1 ^d^	21.7 ^d^	1.4 ^d^				8.9 ^d^			
Yellow grease				2.43 ^fh^	23.24 ^fh^		12.96 ^fh^	44.32 ^fh^	6.97 ^fh^	0.67 ^fh^						

The values of different fatty acids reported by different studies are represented by superscripts ^a^ [88], ^b^ [89], ^c^ [68,90,91,92,93,94]; microalgae species (*Nannochlopsis oculate*) ^d^ [95], ^e^ [66], ^f^ [96], ^g^ [75] and ^h^ [97].

**Table 5 bioengineering-10-00029-t005:** Physicochemical properties of different biofuel feedstocks.

Sources	CP (°C)	CN	OS (mg/100 mL)	SV	IN	AV (mg KOH/g oil)
Soybean oil	0.9	47	16.0	189–195	117–143	0.1–0.2
Canola oil	−3.3	55	44.9	188–193	109–126	0.6–0.8
Olive	-	-	-	184–196	75–94	0.94–2.11
Corn	-	-	-	187–198	103–140	0.1–5.75
Jatropha curcas	5.66	55.43	-	177–189	92–112	15.6–43
Palm oil	14.24	60.21	-	186–209	35–61	6.9–50.8
Rapeseed			-	168–187	94–129	0.2
Sunflower			-	186–194	110–143	0.2–0.5
Camelina	2.5	48.91	-		146.5	0.2
Poultry fat	-	-	-	-	78.8	0.55
Choice white grease	7.0	64	72.0	-	-	-
Inedible tallow	16.0	62	6.2	-	-	-
Yellow grease	6.0	58	2.3	-	-	-
Ultra-low sulfur diesel (ULSD)	−45 to −7	47	-	-	-	-

Cloud point (CP), cetane number (CN), oxidative stability (OS), saponification value (SN), iodine number (IN), and Acid value (AV). Values shaded with green are adopted from [88], blue from a study by [75], and not highlighted text black are from the U.S. Energy Information Administration (EIA), compiled from the U.S. Department of Energy, National Renewable Energy Laboratory, and Renewable Energy Group.

**Table 6 bioengineering-10-00029-t006:** Merits and demerits of using various biodiesel production technologies [27,97,133,135].

Production Technologies	Merits	Demerits
Micro-emulsion	Micro-emulsion is a simple process, a potential solution for solving the problem of vegetable oil viscosity [136]. It is the dispersion of water, oil, and surfactant. Alcohols such as methanol and ethanol are used to lower viscosity, higher alcohols are used as surfactants, and alkyl nitrates are used as cetane improvers [137]. Micro-emulsion is an alternative method that produces biofuel with suitable properties with low energy consumption [138].	Some of the disadvantages of micro-emulsion include high viscosity, poor stability, and volatility. Therefore, pre-treatment technology such as cracking, blending, and hydrodeoxygenation is required to minimize the viscosity and FFAs content before producing biodiesel [138].
Pyrolysis	Pyrolysis is a simple and pollution-free process. The product from pyrolysis has a lower viscosity, flash point, and pour point than petroleum diesel; however, it has equivalent calorific values and a lower value of cetane number. Thus, pyrolyzed vegetable oil has an acceptable amount of sulfur, water, sediment, and copper corrosion values [139]. A study suggested that pyrolytic oil, also known as bio-oil, derived from non-edible feedstock such as Jatropha, Castor, Kusum, Mahua, Neem, and Polanga, has drawn interest to be used as an alternative biofuel. The advantages of using pyrolytic bio-oil are that it is easy to handle, store, and transport and has a high cetane number, low viscosity, and low sulfur quantities [138,140].	The bio-oils derived from edible and non-edible plant seeds are acidic. They are denser than petroleum diesel fuel and thus require a pre-treatment process to remove moisture and neutralize prior to use as an alternative biofuel [138,141]. The disadvantages of pyrolysis include high temperature, expensive apparatus, and low purity due to intolerable amounts of carbon residue and clinker [138,141].
Transesterification	The transesterification process has several advantages over the biodiesel synthesis methods, which include eco-friendly, mild chemical reactions, and are suitable for biodiesel feedstock. It effectively reduces moisture, FFAs, and viscosity during producing biodiesel from non-edible oil [138,142].	The type of catalyst used will determine the conversion efficiency, reusability, cost, and applicability of feedstocks with water and high fatty acid content. The enzymes used during the process are costly, and the reaction is time-consuming [4].
Catalytic distillation	Catalytic distillation is a green reactor technology that integrates chemical reactions and product separation into a single operation. This method simultaneously carries out the chemical reaction and product separation within a single-stage operation. The continuous removal of the product from the reactive section via distillation action can lead to increased product yield and enhanced productivity. Catalytic distillation has several advantages, such as mitigating catalyst hot spots, better temperature control, and improved energy integration due to the conduction of an exothermic chemical reaction in a boiling medium. Recent studies show that catalytic distillation is a novel approach to biodiesel production, which is more efficient and cost-effective [143].	The conversion process and solvent usage for post-treatment depend on catalyst recovery.
Dilution	Dilution is a simple process that results in a reduction in the viscosity and density of vegetable oils. A study revealed that adding 4% ethanol to diesel fuel increases the brake thermal efficiency, brake torque, and power [144]. Another study reported that blending non-edible oil with diesel fuel increases the storability, potential improvement of physical properties, and engine performance. Additionally, dilution reduces poor atomization and difficulty handling by conventional fuel injection systems of compression ignition engines [55].	The issues with blending include the formation of carbon in the engine and incomplete combustion.
Microwave technology	The electromagnetic waves generated in the microwave through electric energy transfer energy directly at the molecular level, allowing quick reaction activity and better energy transfer [135]. The catalyst (homogeneous or heterogeneous) in microwave radiation lowers microwave power usage while keeping the reaction equilibrium and achieving transesterification at very low input power with a very fast conversion rate [53]. The high input power can directly degrade oils into different byproducts. Thus, controlling the radiation level is vital to achieving a complete transesterification reaction.	Removal of the catalyst after the process is needed, and process conversion depends on catalyst activity and is not appropriate for solid feedstocks.
Reactive distillation	Reactive distillation offers new and exciting opportunities for manufacturing fatty acid alkyl esters in the industrial production of biodiesel and specialty chemicals. The processes can be enhanced by heat integration and powered by heterogeneous catalysts to eliminate all conventional catalyst-related operations by efficiently using raw materials and reaction volume. At the same time, reactive distillation offers higher conversion, selectivity, and high energy savings [145]. This method combines the reaction and separation stages in a single unit, thereby reducing the capital cost and increasing heat integration [25]. Overall, this method is applicable with feedstock with high FFAs content, simple process, less use of methanol, and easy to separate product.	However, it requires high energy, and process conversion depends on catalyst efficiency.
Supercritical fluid method	In the supercritical fluid method, the reaction is carried out at supercritical conditions. The mixture becomes homogeneous, where both the esterification of free fatty acids and the transesterification of triglycerides occur without needing a catalyst, making the process suitable for all types of raw materials. The combination of two stages has attracted research interest recently, where simultaneous extraction and reaction from solid matrices are carried out using methanol with supercritical CO_2_ as a co-solvent [25]. This method involves less reaction time, high conversion, and no catalyst required.	This method demands a high cost of apparatus and energy consumption.

**Table 7 bioengineering-10-00029-t007:** Comparison of acid versus alkali-catalyzed transesterification process of biodiesel production, reported by [27].

Transesterification Process	Merits	Demerits
Acid-based catalyzed reaction	Suitable in the presence of high levels of FFA and water. No need for pretreatment. Fewer environmental problems and less toxic effect. Few main processing units.	Slow reaction. High temperature, pressure, and alcohol/oil ratio. Environmental contamination. Required costly equipment.
Alkali-based catalyzed reaction	Low temperature, pressure, and alcohol/oil ratio. High reaction rate. Smaller equipment, good corrosion resistance properties. Low cost of catalyst.	Need of pretreatment. Low ester yields and byproducts without pretreatment. Saponification occurs.

**Table 8 bioengineering-10-00029-t008:** Comparison of homogeneous versus heterogeneous-catalyzed transesterification process of biodiesel production, reported by [27].

Factors	Homogeneous Catalysis	Heterogenous Catalysis
Reaction rate	Fast and high conversion	Moderate conversion
Post-treatment	No recovery of catalyst	Catalysts can be recovered
Processing methodology	Mild reaction and less energy consumption	Continuous operation possible
Process of water and FFA	Sensitive and not suitable	Not sensitive and suitable
Reuse of catalyst	Not possible	Possible
Cost	Comparatively cost-effective than the currently available heterogeneous catalyzed transesterification	Potentially cheaper, high conversion efficiency, and technologically available

## Data Availability

Not applicable. There is no primary data used in this study. However, secondary data that supports this study are available from the corresponding author (D.N.) upon reasonable request.

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
