# Peer review of "Biofuels from Renewable Sources, a Potential Option for Biodiesel Production"

_bioengineering, 2022, doi:10.3390/bioengineering10010029_

Round 1

Reviewer 1 Report

This paper by D. Neupane is a review on biofuels. It is a very useful manuscript with extensive literature research and comprehensive structure. The topic of the study is of interests. It is basically suitable for publication in the journal, however there are some issues to be addressed below.

Major comments

1.     Section 2.1 and Figure 1, the notation should be unified in the section 2 and Figure 1. For example, if author introduces 2.1.4 Green diesel and 2.1.6 Solid biofuels, it should also be shown in Figure 1.

2.     Tables 1 and 2, the titles are too long. Should change to concise title and add footnote.

3.     Table 1, considering the description of this table, it is unclear whether it is necessary to indicate the harvest amount by month.

4.     Lines 385-386, “micro-emulsion requires high temperatures”, is this sentence true? Microemulsions require surfactants, solvents and water and can generally be formed at room temperature.

5.     Table 6, comprehensive comments on catalytic distillation, dilution, microwave technology, reactive distillation, and superfluid method are required.

6.     N2O is the only greenhouse gas among NOx. When discussing the environmental impact of using biodiesel, the focus should be on N2O. Authors should comment on N2O emissions.

Minor comments

1.     Line 78, “bioethanol” is written twice.

2.     Line 115, “Figure 2” should be replaced with “Figure 2”.

3.     Line 122, “Biodiesel, Biogas” should be replaced with “biodiesel, biogas”.

4.     Line 128, “gravimetric” should be deleted.

5.     Line 165, what is (11) after biofuels?

6.     Line 178, “Fatty Acid Methyl Ester (FAME)” should be replaced with “fatty acid methyl ester (FAME).”

7.     Lines 208-209, “Biodiesel produced from certain feedstock has unique properties compered to petroleum diesel.” The relationship between this sentence and the following sentence is unclear.

8.     Line 218, “In the USA, soybean oil is the primary source of biodiesel.” This sentence should be deleted because the same sentence is described in line 214 immediately before.

9.     Line 232, the paper by Gülşen et al. should be added to the references.

10.  Line 288, “Yellow” should be replaced with “yellow”.

11.  Line 315, “also known as fatty acid methyl esters (FAME)” should be replaced with “also known as FAME”.

12.  Line 429, “6.3 Types of Alcohol” should be replaced with “Types of alcohol”.

13.  Lines 536-537, numbers should be subscripted.

Author Response

Dear reviewers,

Thank you very much for your constructive comments and suggestions for the manuscript entitled "Biofuels from renewable sources, a potential option for biodiesel production." Your comments and suggestions help a lot to improve the overall quality of the manuscript. I appreciated them and addressed them as per the suggestions.

Reviewer 1

Comments and Suggestions for Authors

This paper by D. Neupane is a review on biofuels. It is a very useful manuscript with extensive literature research and comprehensive structure. The topic of the study is of interest. It is basically suitable for publication in the journal, however there are some issues to be addressed below.

Response to reviewer 1: Thank you for your comments and suggestions. Please find the attached copy of my response to your concern.

Major comments

  1. Section 2.1 and Figure 1, the notation should be unified in the section 2 and Figure 1. For example, if the author introduces 2.1.4 Green diesel and 2.1.6 Solid biofuels, it should also be shown in Figure 1.

Response: addressed.

  1. Tables 1 and 2, the titles are too long. Should change to concise title and add footnote.

Response: addressed.

  1. Table 1, considering the description of this table, it is unclear whether it is necessary to indicate the harvest amount by month.

Response: You are right. However, it will provide detail information on what amount was produced each month. In some case, for example, cottonseed oil if I just include 0.3 million kg, it is not clear on what happened each month. 

  1. Lines 385-386, “micro-emulsion requires high temperatures”, is this sentence true? Microemulsions require surfactants, solvents and water and can generally be formed at room temperature.

Response: This confusing sentence was removed.

The following sentences were added: Microemulsion-based fuel systems reduce the combustion temperature, which leads to lower emissions of thermal NOx, CO, black smoke, and particulate matter. However, one major problem of using ethanol to formulate a microemulsion system is its lower miscibility with diesel. The immiscibility can be visualized for a wide range of temperatures, particularly at lower temperatures. Furthermore, environmentally benign bio-based non-ionic surfactants and cosurfactant without N and S are of environmental concern.

  1. Table 6, comprehensive comments on catalytic distillation, dilution, microwave technology, reactive distillation, and superfluid method are required.

Response: addressed. Please check table 6.

  1. N2O is the only greenhouse gas among NOx. When discussing the environmental impact of using biodiesel, the focus should be on N2O. Authors should comment on N2O emissions.

     Response: The following paragraph was added on N2O emission.

      Among NOx, nitrous oxide (N2O) is only the greenhouse gas of great environmental concern. It is a substantial anthropogenic greenhouse gas, and agriculture represents its most significant source. The global warming potential of N2O is 298 times that of CO2 [82]. Previous studies on biofuel production systems revealed that emissions of N2O may counterbalance a substantial part of the global warming reduction by fossil fuel displacement [83]. Using optimized crop management, which involves state-of-the-art agricultural technologies coupled with an optimized fertilization regime, and nitrification inhibitors, N2O emissions can significantly be reduced by -135% points (pp) compared to conventional management. However, uncertainties in using statistical N2O emission models and data on non-land use GHG emissions due to biofuel production are significant, which can change the GHG emission reduction by between -152 and 87 pp [84].

Minor comments

  1. Line 78, “bioethanol” is written twice.

Response: removed.

  1. Line 115, “Figure 2” should be replaced with “Figure 2”.

Response: replaced.

  1. Line 122, “Biodiesel, Biogas” should be replaced with “biodiesel, biogas”.

Response: replaced.

  1. Line 128, “gravimetric” should be deleted.

Response: deleted.

  1. Line 165, what is (11) after biofuels?

Response: removed.

  1. Line 178, “Fatty Acid Methyl Ester (FAME)” should be replaced with “fatty acid methyl ester (FAME).”

Response: replaced.

  1. Lines 208-209, "Biodiesel produced from certain feedstock has unique properties compared to petroleum diesel." The relationship between this sentence and the following sentence is unclear.

Response: removed the entire sentence.

  1. Line 218, “In the USA, soybean oil is the primary source of biodiesel.” This sentence should be deleted because the same sentence is described in line 214 immediately before.

Response: deleted.

  1. Line 232, the paper by Gülşen et al. should be added to the references.

Response: added.

  1. Line 288, “Yellow” should be replaced with “yellow”.

Response: replaced.

  1. Line 315, “also known as fatty acid methyl esters (FAME)” should be replaced with “also known as FAME”.

Response: replaced.

  1. Line 429, “6.3 Types of Alcohol” should be replaced with “Types of alcohol”.

Response: replaced.

  1. Lines 536-537, numbers should be subscripted.

Response: subscripted.

Reviewer 2 Report

The review, submitted by D. Neupane to Bioengineering, is focused on biodiesel. The review comes after a long list of reviews on biofuels. It is a problem that interesting reviews are not cited, and other reviews and journals with little relevance to the subject (i.e., biodiesel) are cited (References 4, 12, 13).

For a discussion of the environmental sustainability of biofuels, see Proc. R. Soc. A 476 (2020) 20200351; DOI: 10.1098/rspa.2020.0351

For a discussion on bioenergy and biofuel production, see Mater. Today: Proc. 49 (2022) 510-516; DOI: 10.1016/j.matpr.2021.03.212

For a discussion on the prospective production of biofuel from microalgae, see Biotechnol. Rep. 27 (2020) e00509; DOI: 10.1016/j.btre.2020.e00509

Furthermore, the author should position himself concerning these numerous reviews. How his review is new and original, and how it stands about the literature.

In section 2.1.2. biodiesels are defined as long-chain fatty esters. In figure 1, biodiesels (without s) are listed as 3rd generation biofuels but are not listed as 2nd generation biofuels. However, I could not find long-chain fatty esters related to the 4th and 5th-generation biofuels in references 22 and 49.

Table 3. The oil contents of various feedstocks are missing. I can say that the oil content of waste cooking oil is >95%. The author should at least give an estimate for the other feedstocks.

Section 5. How could long-chain fatty esters be obtained using pyrolysis? Reference 104, cited by the authors, does not deal with pyrolysis. It’s a review on the advancements in the development and characterization of biodiesel and pyrolysis was only mentioned 2 times in the introduction.

Section 6.3. Why did methanol was chosen as alcohol for the transesterification reaction? Price is an issue, but not the main one.

Section 6.5. Seriously? We need a review to highlight that “Reaction time plays a significant role in product conversion. About 99% of conversion is possible at a higher reaction time” lines 442-443.

The same is true for all sections 6.4, 6.6, 6.8 … “Biodiesel formation is greatly affected by the concentration of the catalyst” line 465! It looks like the author has nothing to say, and he lines up clichés without any interest. 6.9 “Adding reactants is crucial to complete the transesterification process and improve FAME production [134]. Lower agitation speed lowers the production formation” lines 475-476.

Section 9: conclusion. Nowhere in the text it was mentioned that the EU ban on new fossil fuel cars from 2035, like California. Again in the 4th and 5th generations of biofuels, I could not find the long-chain fatty esters.

In conclusion, I still wonder what the point of this review is and what I have learned from reading it. Apart from the still very hypothetical fuels of the 5th generation ...  The author should justify why his review is new and necessary and its objectives. In the present form, I would not approve the publication of this review.

Author Response

 Dear reviewer,

Thank you very much for your constructive comments and suggestions for the manuscript entitled "Biofuels from renewable sources, a potential option for biodiesel production." Your comments and suggestions help a lot to improve the overall quality of the manuscript. I appreciated them and addressed them as per the suggestions.

Reviewer 2

Comments and Suggestions for Authors

The review, submitted by D. Neupane to Bioengineering, is focused on biodiesel. The review comes after a long list of reviews on biofuels. It is a problem that interesting reviews are not cited, and other reviews and journals with little relevance to the subject (i.e., biodiesel) are cited (References 4, 12, 13).

For a discussion of the environmental sustainability of biofuels, see Proc. R. Soc. A 476 (2020) 20200351; DOI: 10.1098/rspa.2020.0351

For a discussion on bioenergy and biofuel production, see Mater. Today: Proc. 49 (2022) 510-516; DOI: 10.1016/j.matpr.2021.03.212

For a discussion on the prospective production of biofuel from microalgae, see Biotechnol. Rep. 27 (2020) e00509; DOI: 10.1016/j.btre.2020.e00509

Response to reviewer 2: Thank you for your comments and suggestions. Please find the attached copy of my response to your concern. Thank you for those excellent papers; I have included them in the review, and additional papers from the citations of these papers.

Furthermore, the author should position himself concerning these numerous reviews. How his review is new and original, and how it stands about the literature.

Response: Thank you again for your great concern. This review is a comprehensive one, which is a synthesis of several literatures. It compiles several sections on biofuels and biodiesel production so readers can enjoy many aspects by reading a single review paper. This will garner much interest among readers in biofuel and biodiesel and will be widely cited. Thank you again for your comments and suggestions, which help me improve this manuscript's overall quality.

In section 2.1.2. biodiesels are defined as long-chain fatty esters. In figure 1, biodiesels (without s) are listed as 3rd generation biofuels but are not listed as 2nd generation biofuels. However, I could not find long-chain fatty esters related to the 4th and 5th-generation biofuels in references 22 and 49.

Response:

Figure 1 is revised, and Fourth-generation biofuels section is completely modified as below:

With the application of molecular biology, genetic engineering, and interdisciplinary physicochemical approaches, which include the use of CRISPR/Cas9 with guided RNA for genetic modification in algae [56] to optimize and enhance the yield of biofuel production, the biofuel generated by such process is considered a fourth-generation biofuel. The fourth-generation biofuel production employs genetically modified algae that accumulate high lipid and carbohydrate content to improve biofuel yield [57]. The raw materials used for biofuel production are microalgae, macroalgae, and cyno-bacteria. Cyno-bacteria are non-photosynthetic prokaryotes, and micro and macro algae are eukaryotes [58]. The inactivation of ADP-glucose phosphorylase in a Chlamydomonas starchless mutant led to a 10-fold increase in TAG [59]. Similarly, a modification in the CoA-dependent 1-butanol production pathway into a cyanobacterium, Synechococcus elongatus, can produce butanol from CO2 directly [60].

5th-generation biofuel section is completely removed as per all the reviewer’s suggestion.

 Table 3. The oil contents of various feedstocks are missing. I can say that the oil content of waste cooking oil is >95%. The author should at least give an estimate for the other feedstocks.

Response: Mentioned the oil content for all the feedstocks used in Table 3. However, I am not able to find any articles that shows oil content of waste cooking oil >95%. If you know any articles, I would like to incorporate that as well.

 Section 5. How could long-chain fatty esters be obtained using pyrolysis? Reference 104, cited by the authors, does not deal with pyrolysis. It’s a review on the advancements in the development and characterization of biodiesel and pyrolysis was only mentioned 2 times in the introduction.

Response: Reference 104 was removed, and detail description about pyrolysis was added. Please refer 5.2, and Table 6 as well.

Section 6.3. Why did methanol was chosen as alcohol for the transesterification reaction? Price is an issue, but not the main one.

Response: Addressed. A couple of sentences were added to clarify the reason why it is better or least important.

 Section 6.5. Seriously? We need a review to highlight that “Reaction time plays a significant role in product conversion. About 99% of conversion is possible at a higher reaction time” lines 442-443.

Response: The contradictory sentence was removed.

The same is true for all sections 6.4, 6.6, 6.8 … “Biodiesel formation is greatly affected by the concentration of the catalyst” line 465! It looks like the author has nothing to say, and he lines up clichés without any interest. 6.9 “Adding reactants is crucial to complete the transesterification process and improve FAME production [134]. Lower agitation speed lowers the production formation” lines 475-476.

Response: Contradictory sentences were removed.

6.4 The ratio of alcohol to oil has a significant influence on biodiesel production.

6.6 The transesterification reaction is impacted by temperature, rate of reaction, and product output as the temperature increases.

6.8 Biodiesel formation is greatly affected by the concentration of the catalyst”.

 6.9 Adding reactants is crucial to complete the transesterification process and improve FAME production.

The section is revised as per your comments and suggestion. Please see section 6 Factors affecting biodiesel production.

Section 9: conclusion. Nowhere in the text it was mentioned that the EU ban on new fossil fuel cars from 2035, like California. Again in the 4th and 5th generations of biofuels, I could not find the long-chain fatty esters.

 In conclusion, I still wonder what the point of this review is and what I have learned from reading it. Apart from the still very hypothetical fuels of the 5th generation ...  The author should justify why his review is new and necessary and its objectives. In the present form, I would not approve the publication of this review.

Response: Conclusion part is revised.

Reviewer 3 Report

This manuscript bioengineering-2013397 is a good up-to-date review on the production of biodiesel via the transesterification of triglycerides contained in oils from diverse sources. Description of all key aspects is notable, though there are some aspects that can be modified or completed:

1) The definition of the generations of oil-based biorefineries is complete. However, all biomass of the lignocellulosic type, including biomass poor in lignin, such as hemp, should be included in what is known as second generation biorefinery. This is globally accepted, though there is a trend to select or create biomass poor in lignin for biofuel (bioethanol, biodiesel, green diesel) production, as lignin creates notable hindrances in the processes leading to biofuels. Thus, the definition of the fifth-generation seems not necessary.

2) As for the third and fourth generation biorefineries, it would be fine to distinguish between the usual composition in vegetable oils (terrestial plants) and oils from algae, indicating the main differences.

3) Regarding the catalysts used in transesterification, it would be fine to make two classifications; one for the type of catalyst (acid, basic, enzymatic) and one for the type of system in which such catalysts are applied: biphasic or triphasis systems; homogeneous catalysts trend to work on liquid-liquid systems, while heterogeneous catalysts are usually solid catalysts suspended in a liquid-liquid mixture (so the system is a triphasic L-L-S system).

The author should avoid numbering subsections as short as 2-4 lines; with so small subsections it is enough to use different font from the normal text for the title of the subsection (bold, underlined...).

Author Response

 Dear reviewers,

Thank you very much for your constructive comments and suggestions for the manuscript entitled "Biofuels from renewable sources, a potential option for biodiesel production." Your comments and suggestions help a lot to improve the overall quality of the manuscript. I appreciated them and addressed them as per the suggestions.

 Reviewer 3

Comments and Suggestions for Authors

This manuscript bioengineering-2013397 is a good up-to-date review on the production of biodiesel via the transesterification of triglycerides contained in oils from diverse sources. Description of all key aspects is notable, though there are some aspects that can be modified or completed:

1) The definition of the generations of oil-based biorefineries is complete. However, all biomass of the lignocellulosic type, including biomass poor in lignin, such as hemp, should be included in what is known as second generation biorefinery. This is globally accepted, though there is a trend to select or create biomass poor in lignin for biofuel (bioethanol, biodiesel, green diesel) production, as lignin creates notable hindrances in the processes leading to biofuels. Thus, the definition of the fifth-generation seems not necessary.

Response: Removed definition of the fifth-generation part.

2) As for the third and fourth generation biorefineries, it would be fine to distinguish between the usual composition in vegetable oils (terrestrial plants) and oils from algae, indicating the main differences.

Response: Mentioned now. Please refer to the text.

3) Regarding the catalysts used in transesterification, it would be fine to make two classifications; one for the type of catalyst (acid, basic, enzymatic) and one for the type of system in which such catalysts are applied: biphasic or triphasic systems; homogeneous catalysts trend to work on liquid-liquid systems, while heterogeneous catalysts are usually solid catalysts suspended in a liquid-liquid mixture (so the system is a triphasic L-L-S system).

Response: Some authors described just two types of catalysts such as homogeneous and heterogeneous, and acid, base, and enzymes are sub-categories of two broad categories. I have added a couple of sentences to address your comments. I have also created Tables 7 and 8 to distinguish between acid-based versus alkali-based and homogeneous versus heterogeneous catalysts. Please refer to the text and tables.

The author should avoid numbering subsections as short as 2-4 lines; with so small subsections it is enough to use different font from the normal text for the title of the subsection (bold, underlined...).

Response: addressed.

Reviewer 4 Report

Dear Authors,

I have reviewed the paper "Pulsed electric field ablation of epicardial autonomic ganglia: Computer analysis of monopolar electric field across the tissues involved". The aims of the paper are germane with Bioengineering journal, in this form of article fits with the international scientific standards, some minor flaws are present. The paper is written with a good English level. The contribution of this paper to the scientific knowledge in the present form in my opinion it is acceptable. Considering what is written above and after an accurate critical reading of the paper, I suggest to revise the paper following the suggestions wrote in the file attached.

Author Response

 Dear reviewers,

Thank you very much for your constructive comments and suggestions for the manuscript entitled "Biofuels from renewable sources, a potential option for biodiesel production." Your comments and suggestions help a lot to improve the overall quality of the manuscript. I appreciated them and addressed them as per the suggestions.

Reviewer 1

Comments and Suggestions for Authors

This paper by D. Neupane is a review on biofuels. It is a very useful manuscript with extensive literature research and comprehensive structure. The topic of the study is of interest. It is basically suitable for publication in the journal, however there are some issues to be addressed below.

Response to reviewer 1: Thank you for your comments and suggestions. Please find the attached copy of my response to your concern.

Major comments

  1. Section 2.1 and Figure 1, the notation should be unified in the section 2 and Figure 1. For example, if the author introduces 2.1.4 Green diesel and 2.1.6 Solid biofuels, it should also be shown in Figure 1.

Response: addressed.

  1. Tables 1 and 2, the titles are too long. Should change to concise title and add footnote.

Response: addressed.

  1. Table 1, considering the description of this table, it is unclear whether it is necessary to indicate the harvest amount by month.

Response: You are right. However, it will provide detail information on what amount was produced each month. In some case, for example, cottonseed oil if I just include 0.3 million kg, it is not clear on what happened each month. 

  1. Lines 385-386, “micro-emulsion requires high temperatures”, is this sentence true? Microemulsions require surfactants, solvents and water and can generally be formed at room temperature.

Response: This confusing sentence was removed.

The following sentences were added: Microemulsion-based fuel systems reduce the combustion temperature, which leads to lower emissions of thermal NOx, CO, black smoke, and particulate matter. However, one major problem of using ethanol to formulate a microemulsion system is its lower miscibility with diesel. The immiscibility can be visualized for a wide range of temperatures, particularly at lower temperatures. Furthermore, environmentally benign bio-based non-ionic surfactants and cosurfactant without N and S are of environmental concern.

  1. Table 6, comprehensive comments on catalytic distillation, dilution, microwave technology, reactive distillation, and superfluid method are required.

Response: addressed. Please check table 6.

  1. N2O is the only greenhouse gas among NOx. When discussing the environmental impact of using biodiesel, the focus should be on N2O. Authors should comment on N2O emissions.

     Response: The following paragraph was added on N2O emission.

      Among NOx, nitrous oxide (N2O) is only the greenhouse gas of great environmental concern. It is a substantial anthropogenic greenhouse gas, and agriculture represents its most significant source. The global warming potential of N2O is 298 times that of CO2 [77]. Previous studies on biofuel production systems revealed that emissions of N2O may counterbalance a substantial part of the global warming reduction by fossil fuel displacement [78]. Using optimized crop management, which involves state-of-the-art agricultural technologies coupled with an optimized fertilization regime, and nitrification inhibitors, N2O emissions can significantly be reduced by -135% points (pp) compared to conventional management. However, uncertainties in using statistical N2O emission models and data on non-land use GHG emissions due to biofuel production are significant, which can change the GHG emission reduction by between -152 and 87 pp [79].   

Minor comments

  1. Line 78, “bioethanol” is written twice.

Response: removed.

  1. Line 115, “Figure 2” should be replaced with “Figure 2”.

Response: replaced.

  1. Line 122, “Biodiesel, Biogas” should be replaced with “biodiesel, biogas”.

Response: replaced.

  1. Line 128, “gravimetric” should be deleted.

Response: deleted.

  1. Line 165, what is (11) after biofuels?

Response: removed.

  1. Line 178, “Fatty Acid Methyl Ester (FAME)” should be replaced with “fatty acid methyl ester (FAME).”

Response: replaced.

  1. Lines 208-209, "Biodiesel produced from certain feedstock has unique properties compared to petroleum diesel." The relationship between this sentence and the following sentence is unclear.

Response: removed the entire sentence.

  1. Line 218, “In the USA, soybean oil is the primary source of biodiesel.” This sentence should be deleted because the same sentence is described in line 214 immediately before.

Response: deleted.

  1. Line 232, the paper by Gülşen et al. should be added to the references.

Response: added.

  1. Line 288, “Yellow” should be replaced with “yellow”.

Response: replaced.

  1. Line 315, “also known as fatty acid methyl esters (FAME)” should be replaced with “also known as FAME”.

Response: replaced.

  1. Line 429, “6.3 Types of Alcohol” should be replaced with “Types of alcohol”.

Response: replaced.

  1. Lines 536-537, numbers should be subscripted.

Response: subscripted.

Reviewer 2

Comments and Suggestions for Authors

The review, submitted by D. Neupane to Bioengineering, is focused on biodiesel. The review comes after a long list of reviews on biofuels. It is a problem that interesting reviews are not cited, and other reviews and journals with little relevance to the subject (i.e., biodiesel) are cited (References 4, 12, 13).

For a discussion of the environmental sustainability of biofuels, see Proc. R. Soc. A 476 (2020) 20200351; DOI: 10.1098/rspa.2020.0351

For a discussion on bioenergy and biofuel production, see Mater. Today: Proc. 49 (2022) 510-516; DOI: 10.1016/j.matpr.2021.03.212

For a discussion on the prospective production of biofuel from microalgae, see Biotechnol. Rep. 27 (2020) e00509; DOI: 10.1016/j.btre.2020.e00509

Response to reviewer 2: Thank you for your comments and suggestions. Please find the attached copy of my response to your concern. Thank you for those excellent papers; I have included them in the review.

Furthermore, the author should position himself concerning these numerous reviews. How his review is new and original, and how it stands about the literature.

Response: Thank you again for your great concern. This review is a comprehensive one, which is a synthesis of several literatures. It compiles several sections on biofuels and biodiesel production so readers can enjoy many aspects by reading a single review paper. This will garner much interest among readers in biofuel and biodiesel and will be widely cited. Thank you again for your comments and suggestions, which help me improve this manuscript's overall quality.

In section 2.1.2. biodiesels are defined as long-chain fatty esters. In figure 1, biodiesels (without s) are listed as 3rd generation biofuels but are not listed as 2nd generation biofuels. However, I could not find long-chain fatty esters related to the 4th and 5th-generation biofuels in references 22 and 49.

Response:

Please refer figure 1 (by author).

Fourth-generation biofuels section is completely modified as below:

With the application of molecular biology, genetic engineering, and interdisciplinary physicochemical approaches, which include the use of CRISPR/Cas9 with guided RNA for genetic modification in algae [50] to optimize and enhance the yield of biofuel production, the biofuel generated by such process is considered a fourth-generation biofuel. The fourth-generation biofuel production employs genetically modified algae that accumulate high lipid and carbohydrate content to improve biofuel yield [51]. The raw materials used for biofuel production are microalgae, macroalgae, and cyno-bacteria. Cyno-bacteria are non-photosynthetic prokaryotes, and micro and macro algae are eukaryotes [52]. The inactivation of ADP-glucose phosphorylase in a Chlamydomonas starchless mutant led to a 10-fold increase in TAG [53]. Similarly, a modification in the CoA-dependent 1-butanol production pathway into a cyanobacterium, Synechococcus elongatus can produce butanol from CO2 directly [54].

5th-generation biofuel section is completely removed as per all the reviewer’s suggestion.

 Table 3. The oil contents of various feedstocks are missing. I can say that the oil content of waste cooking oil is >95%. The author should at least give an estimate for the other feedstocks.

Response: Mentioned the oil content for all the feedstocks used in Table 3. However, I am not able to find any articles that shows oil content of waste cooking oil >95%. If you know any articles, I would like to incorporate that as well.

 Section 5. How could long-chain fatty esters be obtained using pyrolysis? Reference 104, cited by the authors, does not deal with pyrolysis. It’s a review on the advancements in the development and characterization of biodiesel and pyrolysis was only mentioned 2 times in the introduction.

Response: Reference 104 was removed, and detail description about pyrolysis was added. Please refer 5.2, and Table 6 as well.

Section 6.3. Why did methanol was chosen as alcohol for the transesterification reaction? Price is an issue, but not the main one.

Response: Addressed. A couple of sentences were added to clarify the reason why it is better or least important.

 Section 6.5. Seriously? We need a review to highlight that “Reaction time plays a significant role in product conversion. About 99% of conversion is possible at a higher reaction time” lines 442-443.

Response: The contradictory sentence was removed.

The same is true for all sections 6.4, 6.6, 6.8 … “Biodiesel formation is greatly affected by the concentration of the catalyst” line 465! It looks like the author has nothing to say, and he lines up clichés without any interest. 6.9 “Adding reactants is crucial to complete the transesterification process and improve FAME production [134]. Lower agitation speed lowers the production formation” lines 475-476.

Response: Contradictory sentences were removed.

6.4 The ratio of alcohol to oil has a significant influence on biodiesel production.

6.6 The transesterification reaction is impacted by temperature, rate of reaction, and product output as the temperature increases.

6.8 Biodiesel formation is greatly affected by the concentration of the catalyst”.

 6.9 Adding reactants is crucial to complete the transesterification process and improve FAME production.

The section is revised as per your comments and suggestion. Please see section 6 Factors affecting biodiesel production.

Section 9: conclusion. Nowhere in the text it was mentioned that the EU ban on new fossil fuel cars from 2035, like California. Again in the 4th and 5th generations of biofuels, I could not find the long-chain fatty esters.

 In conclusion, I still wonder what the point of this review is and what I have learned from reading it. Apart from the still very hypothetical fuels of the 5th generation ...  The author should justify why his review is new and necessary and its objectives. In the present form, I would not approve the publication of this review.

Response: Conclusion part is revised.

Reviewer 3

Comments and Suggestions for Authors

This manuscript bioengineering-2013397 is a good up-to-date review on the production of biodiesel via the transesterification of triglycerides contained in oils from diverse sources. Description of all key aspects is notable, though there are some aspects that can be modified or completed:

1) The definition of the generations of oil-based biorefineries is complete. However, all biomass of the lignocellulosic type, including biomass poor in lignin, such as hemp, should be included in what is known as second generation biorefinery. This is globally accepted, though there is a trend to select or create biomass poor in lignin for biofuel (bioethanol, biodiesel, green diesel) production, as lignin creates notable hindrances in the processes leading to biofuels. Thus, the definition of the fifth-generation seems not necessary.

Response: Removed definition of the fifth-generation part.

2) As for the third and fourth generation biorefineries, it would be fine to distinguish between the usual composition in vegetable oils (terrestrial plants) and oils from algae, indicating the main differences.

Response: Mentioned now. Please refer to the text.

3) Regarding the catalysts used in transesterification, it would be fine to make two classifications; one for the type of catalyst (acid, basic, enzymatic) and one for the type of system in which such catalysts are applied: biphasic or triphasic systems; homogeneous catalysts trend to work on liquid-liquid systems, while heterogeneous catalysts are usually solid catalysts suspended in a liquid-liquid mixture (so the system is a triphasic L-L-S system).

Response: Some authors described just two types of catalysts such as homogeneous and heterogeneous, and acid, base, and enzymes are sub-categories of two broad categories. I have added a couple of sentences to address your comments. I have also created Tables 7 and 8 to distinguish between acid-based versus alkali-based and homogeneous versus heterogeneous catalysts. Please refer to the text and tables.

The author should avoid numbering subsections as short as 2-4 lines; with so small subsections it is enough to use different font from the normal text for the title of the subsection (bold, underlined...).

Response: addressed

Round 2

Reviewer 1 Report

The author has addressed my comments and rivesed the manuscript appropriately. Therefore, this manuscript is suitable for publication.

Reviewer 2 Report

I still wonder what the point of this review is and what I have learned from reading it.

Lines 640-641: “The demands for fossil fuels are gradually increasing due to the improvement in technology (for example, urbanization and improved life standard), which also requires more fuels.”

Lines 645-647: “One of the significant problems in using second-generation vegetable oil is that it lessens engine life if the oil is not refined correctly.”

Lines 654-656: “Many things need to be worked out to make an algal biofuel a commercially viable option to fossil fuel, as the production of biofuels from microalgae is an energy-intensive process [212].”

Again, what I have learned from the review? I was already aware of all these problems which are already published in many journals. Whats’s new. What is the impact of EU ban on new fossil fuel cars from 2035?

I would not approve the publication of this review.